# Lightweight Graph Neural Network Search with Graph Sparsification

## Abstract

Graph Neural Architecture Search (GNAS) has achieved superior performance on various graph-structured tasks. However, existing GNAS studies overlook the applications of GNAS in resource-constraint scenarios. This paper proposes to design a joint graph data and architecture mechanism, which identifies important sub-architectures via the valuable graph data. To search for optimal lightweight Graph Neural Networks (GNNs), we propose Lightweight Graph Neural Architecture Search with Graph SparsIfication and Network Pruning (GASSIP). In particular, GASSIP comprises an operation-pruned architecture search module to enable efficient lightweight GNN search. Meanwhile, we design a novel curriculum graph data sparsification module with an architecture-aware edge-removing difficulty measurement to help select optimal sub-architectures. With the aid of two differentiable masks, we iteratively optimize these two modules to efficiently search for the optimal lightweight architecture. Extensive experiments on five benchmarks demonstrate the effectiveness of GASSIP. Particularly, our method achieves on-par or even higher node classification performance with half or fewer model parameters of searched GNNs and a sparser graph.

## 1 Introduction

Graph data is ubiquitous in our daily life ranging from social networks (Zhang et al., 2022b) and protein interactions (Szklarczyk et al., 2018) to transportation (Wang et al., 2020) and transaction networks (Minakawa et al., 2022). Graph Neural Networks (GNNs) are effective for their ability to model graphs in various downstream tasks such as node classification, link prediction, graph clustering, and graph classification. In order to utilize the graph structure, many GNNs like GCN (Kipf & Welling, 2017), GAT (Veličković et al., 2018), and GraphSAGE (Hamilton et al., 2017) build the neural architecture following the message-passing paradigm (Gilmer et al., 2017): nodes receive and aggregate messages from neighbors and then update their own representations. However, facing diverse graph data and downstream tasks, the manual design of GNNs is laborious. Graph Neural Architecture Search (GNAS) (Gao et al., 2019; Li et al., 2021b; Qin et al., 2021; Guan et al., 2022) tackles this problem and bears fruit for automating the design of high-performance GNNs.

Compared to traditional GNAS, lightweight GNAS offers a wider range of application scenarios by reducing computing resource requirements. However, existing studies have overlooked the significance of this problem. Despite concerns about the accuracy of lightweight models, the lottery ticket hypothesis (Frankle & Carbin, 2018; Chen et al., 2021a) suggests that sub-networks, which remove unimportant parts of the neural network, can achieve comparable performance to the full network. Therefore, the core objective of lightweight GNAS is to efficiently search for the effective sub-architecture corresponding to the high-performance sub-networks. As a result, to achieve the objective of lightweight GNAS, we need to address the two challenges: (1) How to effectively search for optimal GNN sub-architectures? (2) How to efficiently conduct lightweight GNAS?

Regarding the issue of effectiveness, graph neural sub-architectures remain black-box due to neural network complexity, while significant progress has been made in understanding graph data (Zhao et al., 2021; Kang et al., 2020; Rong et al., 2020; Qin et al., 2021). Here, we propose applying graph sparsification to enhance lightweight GNAS, based on the following reason: the underlying assumption of graph sparsification is the existence of a sparse graph that can preserve the accuracy of the full graph for given tasks (Chen et al., 2021a; Rong et al., 2020; Luo et al., 2021; Zheng

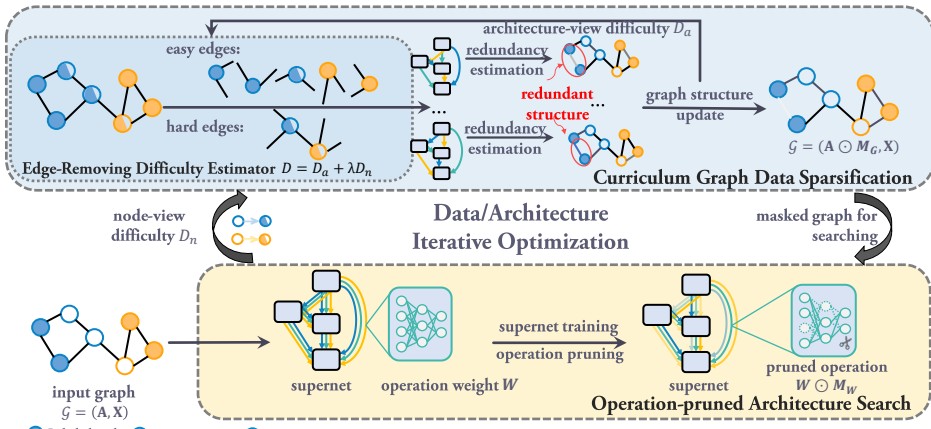

Figure 1: **The iterative training framework of GASSIP.** The graph data and architecture parameters are iteratively optimized. For the operation-pruned architecture search, it first receives the current learned graph structure, then interactively perform supernet training and operation pruning. For the curriculum graph data sparsification, it estimates edge-removing difficulty from node- and architecture-view and updates the graph structure via architecture sampling and sample reweighting.

et al., 2020). Therefore, it is reasonable to infer that the effective sub-architecture plays a crucial role in processing the informative sparse graph. By leveraging the information provided by the sparse graph, we can identify the corresponding sub-architectures. Regarding the issue of efficiency, a straightforward way to realize the lightweight GNAS goal is the first-search-then-prune pipeline, which suffers an efficiency problem since it needs two GNN training sessions. However, there is no existing work discussing how to deal with GNAS and pruning at the same time.

In this paper, we propose Lightweight Graph Neural Architecture Search with Graph SparsIfication and Network Pruning (GASSIP). As shown in Figure 1, GASSIP performs iterative data and architecture optimization through two components: operation-pruned architecture search and curriculum graph data sparsification. The former component helps to construct lightweight GNNs with fewer parameters and the later one helps to search for more effective lightweight GNNs. In particular, we conduct operation pruning with a differentiable operation weight mask to enable identifying important parts of the architecture in the operation-pruned architecture search. Meanwhile, in the curriculum graph data sparsification, we use a differentiable graph structure mask to identify useful edges in graphs and further help search for optimal sub-architectures. To conduct a proper judgment of useful/redundant graph data information, we exploit curriculum learning with an edge-removing difficulty estimator and sample(nodes) reweighting to learn graph structure better.

Meanwhile, our designed joint search and pruning mechanism has comparable accuracy and is far more efficient compared with the first-search-then-pruning pipeline, as shown in experiments. The graph data and operation-pruned architectures are iteratively optimized. Finally, GASSIP generates the optimal sub-architecture and a sparsified graph.

Our contributions are summarized as follows:

- We propose an operation-pruned efficient architecture search method for lightweight GNNs.
- To recognize the redundant parts of graph data and further help identify effective sub-architectures, we design a novel curriculum graph data sparsification algorithm by an architecture-aware edge-removing difficulty measurement.
- We propose an iterative optimization strategy for operation-pruned architecture search and curriculum graph data sparsification, while the graph data sparsification process assists the sub-architecture searching.
- Extensive experiments on five datasets show that our method outperforms vanilla GNNs and GNAS baselines with half or even fewer parameters. For Cora dataset, we improve vanilla GNNs by 2.42% and improve GNAS baselines by 2.11%; the search cost is reduced from 16 minutes for the first-search-then-prune pipeline to within one minute.

## 2 RELATED WORK

However, these works are all manually-designed and are unable to consider graph data and network architecture at the same time.

## 2.1 GRAPH NEURAL ARCHITECTURE SEARCH

The research of Graph Neural Architecture Search (GNAS) has flourished in recent years for automating the GNN architecture design (Zhang et al., 2022a; Xu et al., 2023). We refer the readers to the GNAS survey (Zhang et al., 2021) for details. GraphNAS (Gao et al., 2019) is the first attempt to build the GNN search space and utilizes reinforcement learning to find the optimal architecture. For a more efficient search, many works (Li et al., 2021a; ZHAO et al., 2021; Cai et al., 2021b) adopt the differentiable architecture search algorithm. On a continuous relaxation of the search space, all candidate operations are mixed via architecture parameters, which are updated with operation parameters. Considering the certain noises in graph data, GASSO (Qin et al., 2021) conducts a joint optimization for architecture and graph structure. All previous works only focus on searching for high-performance architectures but overlook searching for a lightweight GNN. As far as we know, the most related work to ours is ALGNN (Cai et al., 2021a). ALGNN searches for lightweight GNNs with multi-objective optimization, but it neglects the vital role of the graph structure, which is important not only for graph representation learning but also for guiding the graph neural architecture search. Aside from the GNAS literature, Yan et al. (2019) also proposed HM-NAS to improve the architecture search performance by loosening the hand-designed heuristics constraint with three hierarchical masks on operations, edges, and network weights. In contrast, our focus is different from HM-NAS as we aim to search for a lightweight GNN considering co-optimizing the graph structure. To achieve goal, we design a novel lightweight graph neural architecture search algorithm that exploits graph data to select optimal lightweight GNNs with a mask on network weights.

## 2.2 GRAPH DATA SPARSIFICATION

Graph data sparsification is to sparsify the graph structure which removes several edges, maintains the information needed for downstream tasks, and allows efficient computations (Zhang et al., 2023; Liu et al., 2023). Some methods rebuild the graph structure through similarity-related kernels based on node embeddings. For example, GNN-Guard (Zhang & Zitnik, 2020) exploits cosine similarity to measure edge weights. Additionally, some algorithms (Zheng et al., 2020; Luo et al., 2021) leverage neural networks to produce intermediate graph structures and then use discrete sampling to refine the graph structure. Furthermore, the direct learning algorithm (Ying et al., 2019; Chen et al., 2021a; Qin et al., 2021) takes the edge weights as parameters by learning a structure mask and removing lower-weight edges. In this paper, we perform graph data sparsification through graph structure learning using the tools from curriculum learning and jointly conduct the architecture search.

## 3 PRELIMINARIES

Let $\mathcal{G} = (\mathbf{A}, \mathbf{X})$ denotes one graph with $N$ nodes $\mathcal{V} = \{\mathcal{V}_L, \mathcal{V}_U\}$, where $\mathcal{V}_L$ is the labeled node set and $\mathcal{V}_U$ is the unlabeled node set, $\mathbf{A} \in \mathbb{R}^{N \times N}$ represents the adjacency matrix (the graph structure) and $\mathbf{X} \in \mathbb{R}^{N \times D_0}$ represents the input node features. $\mathcal{E}$ is the edge set in $\mathcal{G}$.

For a node classification task with $C$ classes, given a GNN $f$, it upgrades the node representations through feature transformation, message propagation, and message aggregation, and outputs node predictions $\mathbf{Z} \in \mathbb{R}^{N \times C}$:

$$\mathbf{Z} = f(\mathbf{A}, \mathbf{X}; \mathbf{W}), \tag{1}$$

where $\mathbf{W}$ denotes network weights. The objective function of the semi-supervised node classification task is the cross-entropy loss between predictions and ground truth labels, denoted as $\mathcal{L}_{clf}$.

## 3.1 DIFFERENTIABLE GRAPH NEURAL ARCHITECTURE SEARCH

The goal of GNAS could be formulated as a bi-level optimization problem (Liu et al., 2018):

$$\begin{aligned} \alpha^* &= \arg\min_{\alpha} \mathcal{L}_{val}(\mathbf{W}^*(\alpha), \alpha) \\ \text{s.t. } \mathbf{W}^*(\alpha) &= \arg\min_{\mathbf{W}} \mathcal{L}_{train}(\mathbf{W}, \alpha), \end{aligned} \tag{2}$$

where $\alpha$ is the architecture parameter indicating the GNN architecture, and $\mathbf{W}$ is the learnable weight parameters for all candidate operations. $\mathbf{W}^*(\alpha)$ is the best weight for current architecture

$\alpha$ based on the training set and $\alpha^*$ is the best architecture according to validation set. Here, we resort to the Differentiable Neural Architecture Search (DARTS) (Liu et al., 2018) algorithm to conduct an efficient search. Considering the discrete nature of architectures, DARTS adopts continuous relaxation of the architecture representation and enables an efficient search process. In particular, DARTS builds the search space with the directed acyclic graph (DAG) (shown as supernet in Figure 1) and each directed edge $(i, j)$ is related to a mixed operation based on the continuous relaxation $\bar{o}^{(i,j)}(\mathbf{x}_i) = \sum_{o \in \mathcal{O}} \frac{\exp(\alpha_o^{(i,j)})}{\sum_{o' \in \mathcal{O}} \exp(\alpha_{o'}^{(i,j)})} o^{(i,j)}(\mathbf{x}_i)$, where $\mathbf{x}_i$ is the input of node $i$ in DAG, $\mathcal{O}$ stands for the candidate operation set (e.g., message-passing layers), and $\alpha$ is the learnable architecture parameter. In the searching phase, weight and architecture parameters are iteratively optimized based on the gradient descent algorithm. In the evaluation phase, the best GNN architecture is induced from mixed operations for each edge in DAG, and the optimal GNN is trained for final evaluation.

Nonetheless, the problem Eq.2 does not produce lightweight GNNs. Next, we introduce the lightweight graph neural architecture search problem and our proposed method.

# 4 LIGHTWEIGHT GNAS

In this section, we introduce our lightweight GNAS algorithm, GASSIP, in detail. First, we formulate the corresponding problem in Sec. 4.1. Then, we describe the curriculum graph data sparsification algorithm in Sec. 4.3. Finally, we introduce the iterative optimization algorithm of the curriculum graph data sparsification and operation-pruned architecture search in Sec. 4.4.

## 4.1 PROBLEM FORMULATION

Here, we introduce two learnable differentiable masks $\mathbf{M}_G, \mathbf{M}_W$ for the graph structure $\mathbf{A}$ and operation weights $\mathbf{W}$ in the supernet. The value of the operation weight mask indicates the importance level of operation weights in the architecture and therefore helps to select important parts in GNN architectures. The trained graph structure mask could identify useful edges and remove redundant ones and thus helps to select important architectures while searching.

The goal of GASSIP could be formulated as the following optimization problem:

$$\alpha^* = \arg\min_{\alpha} \mathcal{L}_{val}(\mathbf{A} \odot \mathbf{M}_G^*, \mathbf{W}^* \odot \mathbf{M}_W^*, \alpha)$$
$$\text{s.t. } \mathbf{W}^*, \mathbf{M}_W^* = \arg\min_{\mathbf{W}, \mathbf{M}_W} \mathcal{L}_{train}(\mathbf{A} \odot \mathbf{M}_G^*, \mathbf{W} \odot \mathbf{M}_W, \alpha),$$
$$\mathbf{M}_G^* = \arg\min_{\mathbf{M}_G} \mathcal{L}_{struct}(\mathbf{A} \odot \mathbf{M}_G, \mathbf{W} \odot \mathbf{M}_W, \alpha),$$

(3)

where $\odot$ denotes the element-wise product operation, $\mathbf{M}_G^*$ indicates the best structure mask based on the current supernet and the structure loss function $\mathcal{L}_{struct}$, $\mathbf{W}^*$ and $\mathbf{M}_W^*$ are optimal for $\alpha$ and current input graph structure $\mathbf{A} \odot \mathbf{M}_G^*$. The target of GASSIP is to find the best discrete architecture according to the architecture parameters $\alpha$, obtain the sparsified graph based on the structure mask $\mathbf{M}_G$ and get the pruned network from the weight mask $\mathbf{M}_W$. In practice, we use sparse matrix-based implementation, which means that $\mathbf{M}_G$ is a $|\mathcal{E}|$-dimensional vector.

## 4.2 OPERATION-PRUNED ARCHITECTURE SEARCH

We leverage network pruning, which reduces the number of trained parameters, to build lightweight GNNs. In contrast with directly building smaller GNNs with fewer hidden channels, building GNNs with reasonable hidden channels and then performing pruning could realize the lightweight goal without compromising accuracy. In GASSIP, we prune the operation in the supernet while searching and name it the operation-pruned architecture search. Specifically, we co-optimize candidate operation weights $\mathbf{W}$ and their learnable weight mask $\mathbf{M}_W = \sigma(\mathbf{S}_W)$ in the searching phase, where $\mathbf{S}_W$ is a trainable parameter and $\sigma$ is a sigmoid function which restricts the mask score between 0 and 1. The differentiable operation weight mask helps to identify important weights in operations.

## 4.3 CURRICULUM GRAPH DATA SPARSIFICATION

Effective sub-architectures could better utilize useful graph information to compete with full architectures. Useful graph data could help to select the most important parts of the GNN architecture

while unsuitable removal of the graph data may mislead the sub-architecture searching process. Here, we exploit graph structure learning to help search for optimal sub-architectures. Besides, we conduct a further graph sparsification step which removes redundant edges after the whole training procedure. The calculation of message-passing layers includes the edge-level message propagation, in which all nodes receive information from their neighbors with $|\mathcal{E}|$ complexity. A sparser graph, compared to a dense graph, has less inference cost because of the decrease in edge-wise message propagation. Hence, eliminating several edges in graph data helps to reduce the model complexity and boosts the model inference efficiency.

In this section, we answer the first question in the Sec. 1 and propose our curriculum graph data sparsification algorithm to guide the lightweight graph neural architecture search in a positive way. A successful graph sparsification could recognize and remove redundant edges in the graph structure. For GNNs, it is natural to identify structure redundancy as edges with low mask scores. However, for GNAS, plenty of architectures are contained in one supernet while different architectures have their own views of redundant information, which is illustrated by observation in Appendix B.

**Structure Redundancy Estimation.** In order to estimate the graph structure redundancy, we exploit structure learning and formulate the graph structure mask $\mathbf{M}_G$ with the sigmoid function $\sigma$:

$$\mathbf{M}_G = \sigma(\mathbf{S}_G - \gamma), \tag{4}$$

where $\mathbf{S}_G \in \mathbb{R}^{N \times N}$ is a learnable mask score parameter and $\gamma$ is a learnable mask threshold parameter which helps to control the graph data sparsity. The number of non-zero elements in $\mathbf{S}_G$ equals $|\mathcal{E}|$. The sigmoid function restricts the graph structure mask score into $(0, 1)$. Smaller structure mask scores indicate that corresponding edges are more likely to be redundant. The structure mask is differentiable and updated through the calculated gradient of the loss function $\mathcal{L}_{struct}$.

Intuitively, if an edge is redundant, it would be regarded as a redundant one no matter what the architecture is. If the updated gradients are consistent under several architectures, we have more confidence to update the structure mask score. Considering the Observation, we propose to leverage backward gradients on different architectures to formulate the structure mask update confidence. In particular, we first sample top-$K$ architectures $\{a_1, a_2, ..., a_K\}$ from the supernet according to the product of the candidate operation probability in each layer:

$$a \sim P_K(\mathcal{O}, \alpha). \tag{5}$$

We calculate the backward gradient $\nabla^{a_i}_{\mathbf{S}_G} = \nabla_{\mathbf{S}_G} \mathcal{L}_{struct}\big(f_{a_i}(\mathbf{A} \odot \mathbf{M}_G, \mathbf{X})\big)$ for each sampled architecture $\{a_i, i = 1, 2, ..., K\}$. Then, we exploit the standard deviation of $\nabla^{a_i}_{\mathbf{S}_G}$ to construct the structure mask update confidence $\mathrm{std}(\nabla^a_{\mathbf{S}_G})$. The final update for the structure mask is formulated as:

$$\nabla_{\mathbf{S}_G} = \frac{\sum_{i=1}^{K} \nabla^{a_i}_{\mathbf{S}_G}}{K \, \mathrm{std}(\nabla^a_{\mathbf{S}_G})}, \quad \mathbf{S}_G \leftarrow \mathbf{S}_G - \eta \nabla_{\mathbf{S}_G}, \tag{6}$$

$$\nabla_{\gamma} = \frac{\sum_{i=1}^{K} \nabla^{a_i}_{\gamma}}{K}, \quad \gamma \leftarrow \gamma - \eta \nabla_{\gamma}. \tag{7}$$

**Curriculum Design.** Some redundant edges are easier to recognize than others. For example, if several architectures have different judgments of one edge's redundancy, it is hard to decide whether this edge should be removed or not. For GNAS, false structure removal in the early stage of searching may misguide the search process. As a result, we introduce curriculum learning into the graph sparsification process based on the architecture-aware edge-removing difficulty measurement and the sample re-weighting strategy. Our method belongs to a more general definition of curriculum learning in which we schedule the training process by softly reweighting and selecting sample nodes rather than directly controlling the node difficulty (Wang et al., 2021).

Specifically, we evaluate the architecture-aware edge-removing difficulty from two views: the architecture view and the node view. From the architecture view, if several architectures have disparate judgments of mask update, the corresponding edge moving should be more difficult. For edge $e_{ij}$ between node $i$ and node $j$, the edge-removing difficulty under the architecture-view is defined as

$$\mathcal{D}_a(e_{ij}) = \mathrm{std}(\nabla^a_{\mathbf{S}_{G,ij}}), \tag{8}$$

where $\mathrm{std}$ indicates the standard deviation. It is worth mentioning that $\mathcal{D}_a(e_{ij})$ has already been calculated in the structure redundancy estimation step, which could be saved in memory without repeating the calculation.

From the node view, edges that link similar nodes are harder to remove and nodes with a lower information-to-noise ratio have more difficult edges. Here, we measure the information-to-noise ratio with label divergence. Therefore, the node-view edge-removing difficulty is evaluated as:

$$\mathcal{D}_n(e_{ij}) = f_{cos}(\mathbf{z}_i, \mathbf{z}_j) + \lambda_1 \frac{\sum_{j \in \mathcal{N}_i} I(\bar{y}_j \neq \bar{y}_i)}{|\mathcal{N}_i|}, \tag{9}$$

where $\lambda_1$ is a hyper-parameter balancing the node-view difficulty, $\mathcal{N}_i$ denotes neighbors of node $i$. $I()$ is the 0-1 indicator function and $f_{cos}$ represents the cosine similarity function. $\mathbf{z}_i$ stands for the final representation of node $i$ calculated in the architecture parameter training phase. $\hat{y}_i$ represents the predicted label and $\bar{y}_i$ is the pseudo-label assigned based on predictions for the output $\mathbf{z}_i$:

$$\bar{y}_i = \begin{cases} \hat{y}_i, & i \in \mathcal{V}_U \\ y_i, & i \in \mathcal{V}_L. \end{cases} \tag{10}$$

Considering the inseparable nature of edges and the ease of usage of nodes in the loss function, we build the node difficulty based on the architecture-aware edge-removing difficulty. We use the sample reweighting strategy during the structure mask training based on the node difficulty.

$$\mathcal{D}(e_{ij}) = \mathcal{D}_a(e_{ij}) + \lambda_2 \mathcal{D}_n(e_{ij}) \tag{11}$$

$$\mathcal{D}(i) = \frac{\sum_{j \in \mathcal{N}_i} \mathcal{D}(e_{ij})}{|\mathcal{N}_i|}, \tag{12}$$

where $\lambda_2$ is a hyper-parameter. In this way, the node difficulty is defined as the average edge-removing difficulty for all its neighbors.

Following the idea of Hard Example Mining (Shrivastava et al., 2016), we regard difficult edges are more informative and need to be weighted more in training. We assign nodes with higher node/edge-removing difficulty higher sample weights. The node weight is calculated as

$$\theta_i = \mathrm{softmax}(\mathcal{D}(i)), \ i \in \mathcal{V} \tag{13}$$

Based on node weights $\mathbf{v}$, the loss function of graph sparsification for sampled architecture $a$ is

$$\mathcal{L}_{struct} = \sum_{i \in \mathcal{V}_L} \theta_i \big( \mathcal{L}_{clf}(f_a(\mathbf{A} \odot \mathbf{M}_G, \mathbf{X}), \bar{y}_i) + \beta \mathcal{L}_{ent}(\mathbf{M}_G) \big), \tag{14}$$

where $\mathcal{L}_{clf}$ is the classification loss based on the assigned pseudo-labels. $\mathcal{L}_{ent}$ is the mean entropy of each non-zero element in $\mathbf{M}_G$, which forces the mask score to be close to 0 or 1. $\beta$ is a hyper-parameter balancing the classification and entropy loss.

The overall curriculum graph data sparsification algorithm is summarized in Algorithm 1. In line 1, pseudo-labels are assigned based on the supernet predictions. Then the node weights in $\mathcal{L}_{struct}$ are updated via edge-removing difficulty calculation in Line 2. In Lines 3-7, $K$ architectures are sampled from the supernet, structural gradients are calculated and the structure mask is updated.

---

**Algorithm 1** Curriculum Graph Data Sparsification.

---

**Input:** The graph data $\mathcal{G}(\mathbf{A}, \mathbf{X})$, candidate operations $\mathcal{O}$, architecture parameters $\alpha$
1 ; **Output:** The structure mask $\mathbf{M}_G$.
2 Assign pseudo-labels $\bar{\mathbf{y}}$ as shown in Eq. 10;
3 Update edge difficulty and assign node weight $\mathbf{v}$ in Eq. 13;
4 Sample $K$ architectures $\{a_1, a_2, ..., a_K\}$ from the supernet according to Eq. 5;
5 **for** $i$ *in* $\{1, 2, ..., K\}$ **do**
6 | Obtain $\nabla_{\mathbf{S}_G}^{a_i}$;
7 Calculate structure mask update confidence $\mathrm{std}(\nabla_{\mathbf{S}_G}^a)$;
8 Update the structure mask $\mathbf{M}_G$ based on Eq. 6 and Eq. 7;
9 Return the structure mask $\mathbf{M}_G$.

---

## 4.4 AN ITERATIVE OPTIMIZATION APPROACH

In this section, we introduce the solution to the second question in the introduction and solve the optimization problem in Eq. 3 in an iterative manner.

Since the informative continuous graph structure helps to select proper operations from the search space while redundant graph data (e.g., noise edges) will deteriorate the architecture search result, we iteratively perform graph sparsification and architecture search optimization. Using the valuable graph data, we pinpoint key components of the GNN for both operations and weights. Furthermore, the introduction of two trainable masks in Eq. 3 enables us to efficiently select useful graph structures and essential parts of the architecture. Fully differentiable parameters, according to DARTS algorithms, can cut the search time of lightweight GNNs from several hours (Cai et al., 2021a) to minutes (shown in Sec. 5.3).

---

**Algorithm 2** The Detailed Algorithm of GASSIP.

---

**Input:** The graph data $\mathcal{G}(\mathbf{A}, \mathbf{X})$, candidate operation set $\mathcal{O}$, training epoch number $T$, warm up epoch number $r$

**Output:** The sparsified graph $\mathcal{G}_{sp}(\mathbf{A} \odot \bar{\mathbf{M}}_G, \mathbf{X})$, optimal lightweight architecture $f_a(\mathcal{G}_{sp}; \mathbf{W} \odot \bar{\mathbf{M}}_W)$.

1 **for** $t \leftarrow 1$ *to* $T$ **do**
2     Update candidate operation weights $\mathbf{W}$ and their masks $\mathbf{M}_W$;
3     **if** $t < r$ **then**
4         continue;
5     Training graph structure mask $\mathbf{M}_G$ following Algorithm 1;
6     Update architecture parameters $\alpha$;
7 Get the binarized structure mask $\bar{\mathbf{M}}_G$ and the binarized weight mask $\bar{\mathbf{M}}_w$;
8 Induce the optimal GNN architecture $a$;
9 Return the sparsified graph $\mathcal{G}_{sp}(\mathbf{A} \odot \bar{\mathbf{M}}_G, \mathbf{X})$ and the optimal lightweight architecture $f_a(\mathcal{G}_{sp}; \mathbf{W} \odot \bar{\mathbf{M}}_W)$.

---

**Training Procedure.** We summarize the whole training procedure in Algorithm 2. Line 1-6 provides the detailed training process of GASSIP. For the first $r$ warm-up epochs, only candidate operation weights and their masks are updated. Then, the operation weights/masks, structure masks, and architecture parameters are iteratively optimized by calculating the gradient descending of objectives in Eq. 3. In practice, the pruning mask becomes quite sparse after several iterations. Therefore, the pruning is mostly sparse matrix multiplication, which is more efficient compared with dense matrix multiplication.

After finishing training, the continuous graph structure mask and operation weight mask are binarized to perform graph sparsification and operation pruning in Line 7. In detail, we initialize binarized structure mask $\bar{\mathbf{M}}_G = \mathbf{M}_G$ and remove edges that have mask values lower than the threshold $\gamma$: $\bar{\mathbf{M}}_{G,ij} = 0$, if $\mathbf{M}_{G,ij} < \gamma$. Meanwhile, to formulate the binarized weight mask $\bar{\mathbf{M}}_W$, we force the operation weight mask values that have non-positive values to zero and weights that have positive mask scores to one. The zero elements will not be trained during the evaluation phase.

At last, the final evaluation is conducted based on the sparsified graph $\mathcal{G}_{sp}$ and the induced pruned architectures $f_a(\mathcal{G}_{sp}; \mathbf{W} \odot \bar{\mathbf{M}}_W)$.

## 5 EXPERIMENTS

In this section, we conduct experiments to demonstrate the effectiveness and efficiency of the proposed algorithm, GASSIP. We also display ablation studies of different components in GASSIP. In addition, the experimental settings, sensitivity analysis for hyper-parameters and the comparison of GASSIP with other graph learning and lightweight GNN methods are deferred in the Appendix.

### 5.1 EXPERIMENTAL RESULTS

**Analysis of Model Accuracy.** We compared GASSIP with vanilla GNNs and automated baselines on the node classification task on five datasets in Table 1. The test accuracy (mean±std) is reported over 100 runs under different random seeds. We find that our proposed algorithm outperforms other baselines in all five datasets. Meanwhile, we can observe that the stds are relatively small, therefore the searched result is not sensitive to the choice of random seed. Among all baselines, only

Table 1: Experimental results for node classification. The test accuracy is averaged for 100 runs (mean±std) using different seeds. OOM means out-of-memory. The best results are in **bold**.

| | Method | Cora | CiteSeer | PubMed | Physics | Ogbn-Arxiv |
|---|---|---|---|---|---|---|
| Vanilla GNNs | GCN (Kipf & Welling, 2017) | 80.93±0.67 | 70.39±0.66 | 79.37±0.39 | 97.43±0.12 | 70.57±0.41 |
| | GAT (Veličković et al., 2018) | 80.78±0.93 | 67.40±1.26 | 78.46±0.31 | 97.76±0.11 | 69.40±0.35 |
| | ARMA (Bianchi et al., 2021) | 81.18±0.62 | 69.31±0.70 | 78.51±0.38 | 96.34±0.08 | 70.79±0.36 |
| Graph Sparsification | DropEdge (Rong et al., 2020) | 82.42±0.65 | 70.45±0.73 | 77.51±0.74 | 96.67±0.19 | 69.33±0.36 |
| | NeuralSparse (Zheng et al., 2020) | 81.14±0.70 | 70.64±0.42 | 78.12±0.31 | 97.86±0.45 | OOM |
| | PTDNet (Luo et al., 2021) | 82.42±0.65 | 70.45±0.73 | 77.51±0.74 | 96.47±0.38 | OOM |
| GNAS | DARTS (Liu et al., 2018) | 81.65±0.48 | 70.00±0.94 | 79.42±0.36 | 98.28±0.07 | 70.58±0.25 |
| | GraphNAS (Gao et al., 2019) | 81.33±0.84 | 70.92±0.61 | 78.87±0.61 | 97.45±0.06 | OOM |
| | GASSO (Qin et al., 2021) | 81.09±0.91 | 68.20±1.09 | 78.15±0.59 | 98.06±0.11 | 70.52±0.31 |
| | GUASS (Guan et al., 2022) | 82.05±0.21 | 70.80±0.41 | 79.48±0.16 | 96.76±0.08 | **71.85±0.41** |
| Ours | GASSIP | **83.20±0.42** | **71.41±0.57** | **79.50±0.30** | **98.46±0.06** | 71.30±0.23 |

(a) Cora   (b) CiteSeer   (c) Physics

Figure 2: Scatter plots showing the relationship between the total number of model parameters and node classification performance on (a) Cora, (b) CiteSeer, and (c) Physics. Methods with ∗ are able to perform graph sparsification. Scatters in the upper left show higher classification performance with lower parameter counts.

DropEdge and GASSO are able to conduct graph sparsification/graph structure learning. DropEdge surpasses the automated baselines in some scenarios, which proves the possible performance improvements of removing edge. In comparison, GASSIP selects removing edges with curriculum sparsification jointly with architecture search rather than random sampling. Compared with GASSO, on the one hand, GASSO directly uses the supernet performance as the classification results without inducing an optimal architecture, which hinders its application in memory-limited scenarios. On the other hand, our method further conducts an edge deleting step after the graph structure learning and is able to perform operation pruning, which makes the searched GNNs more lightweight. Meanwhile, GASSIP achieves better performance than GUASS on smaller graphs, but GUASS could handle graphs with more nodes and edges (Ogbn-Arxiv) as it is specially developed for large-scale datasets. However, our trained model is more lightweight and therefore can be applied in scenarios where computational resources are limited, which is not applicable to GUASS.

**Analysis of Model Parameters.** We further visualized the relationship between model parameter counts and classification test accuracy in scatter plots shown in Figure 2. Except for manually-designed GNNs (GCN, GAT, DropEdge) and GNAS methods (DARTS, GraphNAS), we also compare with an iteratively magnitude-based pruning (IMP) method on GCN (Chen et al., 2021a) and the unified GNN sparsification (UGS) framework (Chen et al., 2021a). IMP iteratively removes $p_1\%$ (we set $p_1 = 20\%$) weights and retrains GCN from rewinding weights. UGS simultaneously prunes the graph structure and the model weights also in an iteratively magnitude-based pruning way. We set the iterative edge removing probability $p_2 = 5\%$. We report the best test performance of IMP and UGS based on the validation performance. The hidden size of various baselines is kept the same for each dataset to make a fair comparison.

As shown in Figure 2, GASSIP achieves higher performance with fewer parameter counts. For the Cora dataset, GASSIP reserves only $50\%$ parameters compared with GCN and $13\%$ compared with GAT. For CiteSeer, our method has $8\%$ parameter counts compared with GAT and $15\%$ compared with ARMA. For Physics, the proposed method keeps only $6\%$ parameters compared to GAT. Among all baselines, only DropEdge, UGS, and GASSIP (with ∗ in Figure 2) could generate sparsified graph. DropEdge needs to load the whole graph in memory to perform edge sampling in each GNN layer. As a result, only UGS and GASSIP have the potential to reduce the inference cost from the edge-level message propagation calculation.

## 5.2 ABLATION STUDY

To get a better understanding of the functional components in GASSIP, we further conduct ablation studies on operation pruning and the curriculum graph sparsification parts. Figure 3 shows bar plots of the test accuracy on Cora and Physics. We evaluate the performance under the same search/training hyper-parameters and report the average accuracy over 100 runs. We compare our method with three variants: *w/o op prn* means to search without pruning operations and only perform curriculum graph data sparsification, *w/o sp* stands for searching architectures without the

curriculum graph data sparsification and only conduct operation pruning, *w/o cur* indicates search architectures with the graph data sparsification part but without the curriculum scheduler.

By comparing GASSIP with its *w/o sp* variant in light green, we could find that GASSIP gains performance improvement from the curriculum graph sparsification part largely. This phenomenon shows that the graph sparsification component leads the operation-pruned architecture search in a positive way and further substantiates the effectiveness of leveraging data to search optimal sub-architectures. Within the curriculum graph sparsification part, performing graph sparsification (graph structure learning) with the curriculum scheduler (*w/o op prn*) behaves better than without it (*w/o cur*). Therefore, the curriculum scheduler helps to learn the graph structure mask better. Besides, the iterative optimization of graph data and operation-pruned architecture works well in gaining performance improvement.

To further illustrate the effectiveness of graph sparsification in our method, we add a new ablation study to substitute our graph sparsification algorithm with DropEdge (Rong et al., 2020), which conducts random edge dropping in the differentiable architecture search process. The classification accuracy on Cora is 79.42±0.63 (DARTS 81.65±0.48, ours 83.20±0.42). This result shows that poorly designed edge removal may be harmful to architecture search.

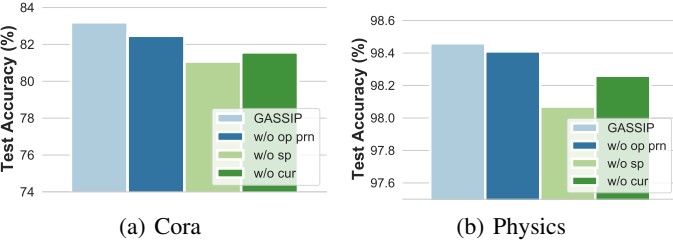

(a) Cora            (b) Physics

Figure 3: Ablation study for GASSIP under scenarios of without operation pruning (*w/o op prn*), without graph data sparsification (*w/o sp*), without curriculum scheduler (*w/o cur*).

## 5.3 SEARCH EFFICIENCY

We compare the search efficiency of GNAS methods in Table 2 and provide the training time cost of searched GNNs in the Appendix. Based on the differentiable architecture search algorithm, GASSIP is more efficient than GraphNAS, which searches architectures with reinforcement learning. The DARTS+UGS baseline represents the first-search-then-prune method which first searches architectures and then conducts network pruning and graph data sparsification. It is loaded with heavy searching, pruning, and retraining costs, which is far less efficient than GASSIP.

Table 2: Searching time cost for GNAS methods.

| Methods | DARTS | DART+UGS | GASSO | GraphNAS | GASSIP |
|---|---|---|---|---|---|
| Search Time (min) | 0.55 | 15.79 | 0.62 | 223.80 | 0.98 |

## 6 CONCLUSION AND LIMITATIONS

In this paper, we propose an efficient lightweight graph neural architecture search algorithm, GASSIP. It iteratively optimizes graph data and architecture through curriculum graph sparsification and operation-pruned architecture search. Our method can reduce the inference cost of searched GNNs at the architecture level by reducing the model parameter counts, and at the data level by eliminating redundant edges. To the best of our knowledge, this is the first work to search for lightweight GNN considering both data and architecture.

**Limitations.** The main purpose of this paper is to search for a lightweight GNN (*i.e.*, lightweight GNN design) that offers a wider range of application scenarios (*e.g.*, edge computing) by limited computational resource requirements. Therefore, the current implementation of GASSIP has difficulty to be integrated with graphs with billions of nodes, This difficulty of scalability commonly hinders both graph sparsification and current GNAS research in applications with contained resources without a specifically designed sampling strategy.

**Future works.** Our future works include evaluating GASSIP on other large-scale graphs, providing a theoretical analysis of the convergence of our iterative optimization algorithm, and developing a unified benchmark for lightweight GNAS.

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

# Appendix: Lightweight Graph Neural Architecture Search with Joint Graph Sparsification and Network Pruning

## A    ADDITIONAL RELATED WORK

### A.1    LIGHTWEIGHT GRAPH NEURAL NETWORKS

The key to building lightweight neural networks is reducing the model parameters and complexity, which further enables neural networks to be deployed to mobile terminals. GNN computation acceleration (Chen et al., 2021b; Abadal et al., 2021; Fu et al., 2022) poses a faster computation for GNNs. Knowledge distillation (Jing et al., 2021; Joshi et al., 2022) follows a teacher–student learning paradigm and transfers knowledge from resource-intensive teacher models to resource-efficient students but keeps performance. Network pruning enables more zero elements in weight matrices. As a result, pruned networks have quicker forward passes for not requiring many floating-point multiplications. For example, (Chen et al., 2021a; You et al., 2021) leverage the iterative magnitude-based pruning and (Liu et al., 2022) uses the gradual magnitude pruning to prune model weights.

## B    OBSERVATION IN CURRICULUM GRAPH DATA SPARSIFICATION

***Observation:*** *Different architectures have their own views of redundant information.*

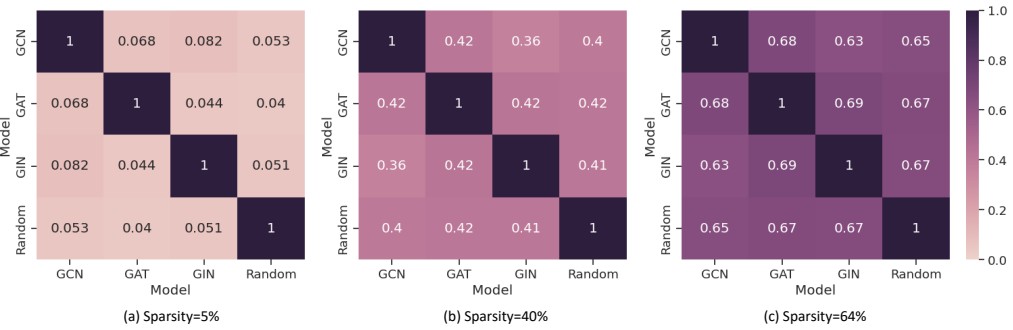

Figure 4: Overlaps of removed edges for GCN, GAT, GIN, and Random under diffident graph data sparsification.

We first train structure masks for three manual-designed GNNs: GCN, GAT, and GIN. Then, we remove edges with the lowest mask scores, and the remove ratio is controlled by a sparsity parameter $p\%$. We calculate the overlaps in the removed edges $s(\mathbf{M}_1, \mathbf{M}_2) = \frac{(\mathbf{A}-\mathbf{M}_1) \cap (\mathbf{A}-\mathbf{M}_2)}{p\%|\mathcal{E}|}$, where $\mathbf{M}_1, \mathbf{M}_2$ are trained binarized structure masks under different manual-designed GNNs. The results are shown in Figure 4. We could see that the removal edges are diverse especially when the number of removal edges is small, which indicates the difference in architectures' judgment for the structure redundancy. We also add a baseline Random which randomly removes $p\%$ edges to demonstrate the low similarities between various masks.

**Dataset.** We evaluate the node classification performance on 5 datasets: Cora, CiteSeer, PubMed, Physics, and Ogbn-Arxiv. The statistics of all datasets are shown in Table 3. The first three datasets follow a traditional semi-node classification train-valid-test split. Physics and Ogbn-Arxiv represent large datasets where we randomly split train:valid:test=50%:25%:25% for Physics and follow the default setting for Ogbn-Arxiv.

**Baselines.** We compare our method with representative hand-crafted GNNs: GCN (Kipf & Welling, 2017), GAT (Veličković et al., 2018), ARMA (Bianchi et al., 2021), and three graph sparsification methods: DropEdge (Rong et al., 2020), PTDNet (Luo et al., 2021), and NeuralSparse (Zheng et al., 2020). We also compare with representative GNAS baselines, including DARTS (Liu et al., 2018), GraphNAS (Gao et al., 2019), GASSO (Qin et al., 2021), and GUASS (Guan et al., 2022). We use GASSO as the representative of GNAS with structure learning.

Table 3: Dataset statistics.

| DATASET | #NODES | #EDGES | #FEATURES | #CLASSES |
|---------|--------|--------|-----------|----------|
| CORA | 2,708 | 10,556 | 1,433 | 7 |
| CITESEER | 3,327 | 9,104 | 3,703 | 6 |
| PUBMED | 19,717 | 88,648 | 500 | 3 |
| PHYSICS | 34,493 | 495,924 | 8,415 | 5 |
| OGBN-ARXIV | 169,343 | 1,166,243 | 128 | 40 |

**Implementation Details.** For GASSIP, we set the number of layers as 2 for CiteSeer, Cora, PubMed, Physics, and 3 for Ogbn-Arxiv. In building search space, we adopt *GCNConv* (Kipf & Welling, 2017), *GATConv* (Veličković et al., 2018), *SAGEConv* (Hamilton et al., 2017), *ArmaConv* (Bianchi et al., 2021), and *Linear* as candidate operations. Due to the memory limit, the search space is narrowed down to *GCNConv,GATConv*, *ArmaConv*, and *Linear* for Physics and Ogbn-Arxiv. The supernet is constructed as a sequence of layers, We set batch normalization (only for Physics and Ogbn-Arxiv) and dropout before each layer and use ReLU as the activation function.

**Detailed Hyper-parameters.** For vanilla GNNs, we follow the hyper-parameters in the original paper except tuning hyper-parameters like hidden channels in $\{16, 64, 128, 256\}$ and dropout in $\{0.5, 0.6, 0.8\}$. For GNAS methods, we use the Adam optimizer to learn parameters. We retrain for 100 runs on their searched optimal architectures to make a fair comparison of the architecture performance. For GASSIP, we fix the number of sampled architectures as $K = 2$, entropy loss the coefficient in curriculum graph data sparsification as $\beta = 0.001$, and the edge-removing difficulty hyper-parameters $\lambda_1 = 1, \lambda_2 = 1$. The supernet training epoch is fixed to 250 and the warm-up epoch number is set as $r = 10$.

## C  SEARCHED ARCHITECTURES IN DETAIL

We follow the literature of graph NAS and construct the supernet as in GASSO and AutoAttend. Here, we provide the searched GNN architecture by GASSIP in Table 4.

Table 4: Search Architectures by GASSIP.

| Dataset | Searched Architecture |
|---------|----------------------|
| Cora | *GCNConv‖SAGEConv* |
| CiteSeer | *GCNConv‖SAGEConv* |
| PubMed | *GCNConv‖ArmaConv* |
| Physics | *Linear‖GCNConv* |
| Ogbn-Arxiv | *GCNConv‖ArmaConv‖GATConv* |

## D  SENSITIVITY ANALYSIS

**hyper-parameters $\lambda_1$ and $\lambda_2$.** We further conduct a sensitivity analysis for curriculum learning hyper-parameters $\lambda_1$ and $\lambda_2$ in Figure 5. Larger $\lambda_1$ indicates that the label divergence is more important in difficulty calculation in our curriculum algorithm while larger $\lambda_2$ suggests that nodes similarity matters more. In the edge-removing difficulty measurement, with these two node difficulty terms ($\lambda_1 > 0, \lambda_2 > 0$) have greater performance than without them ($\lambda = 0$), which further illustrates the difficulty measurement in our curriculum algorithm is reasonable.

## E  RETRAINING EFFICIENCY

We show the (re-)training time cost (s) of searched GNNs in Table 5. For GNAS methods, we report the retraining time of searched GNNs. For manual-designed GNNs, we directly report the training time. For a fair comparison, we fix training epochs as 300 and report the averaged training time over

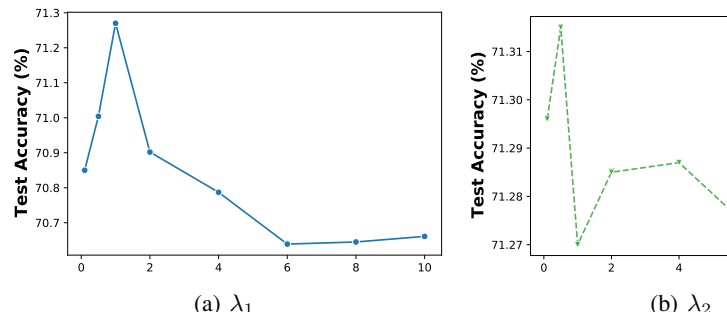

Figure 5: Lineplots for hyper-parameters (a) $\lambda_1$ (fix $\lambda_2 = 1$) and (b) $\lambda_2$ (fix $\lambda_1 = 1$).

100 runs. Results on three datasets illustrate that the training time model searched by GASSIP is the least compared to all baselines, which indicates the efficiency of lightweight GNNs searched by GASSIP.

Table 5: (Re-)training time (s) for searched GNNs averaged over 100 runs.

| Methods | GCN | GAT | ARMA | DropEdge | DARTS | GraphNAS | GASSIP |
|---|---|---|---|---|---|---|---|
| Cora | 2.53 | 3.59 | 2.56 | 2.61 | 2.09 | 3.57 | 2.19 |
| CiteSeer | 2.73 | 3.73 | 3.17 | 2.85 | 3.40 | 3.98 | 2.51 |
| Physics | 11.19 | 70.67 | 79.96 | 13.15 | 10.47 | 64.81 | 5.55 |

**hyper-parameters $K$.** We conduct a sensitivity analysis experiment for hyper-parameter $K$ and the results are shown in Table 6. As the sampled architecture number $K$ in graph structure redundancy estimation gets larger, the classification performance first increases and then drops. A smaller $K$ indicates only a few architectures that are most likely to be selected by NAS are sampled to evaluate the graph structure redundancy. When $K = 1$, only the top-1 architecture induced by current $\alpha$ is sampled. However, it may be not enough to estimate the redundancy and leads to poor graph sparsification. A larger $K$ includes more architectures in the redundancy estimation, but it may contain a lot more architectures that are unlikely to be selected in the search phase and causes poor classification results. As a result, we choose $K$ to be 2 in our experiment. We will add this sensitivity analysis in our revision.

Table 6: Sensitivity analysis for sampled architecture number $K$.

| K | 1 | 2 | 4 | 8 |
|---|---|---|---|---|
| Cora | 81.62±0.67 | 83.20±0.44 | 82.11±0.49 | 81.97±0.41 |
| CiteSeer | 69.42±0.69 | 71.42±0.56 | 70.58±0.48 | 70.23±0.48 |
| Physics | 98.34±0.04 | 98.46±0.06 | 98.29±0.01 | 98.30±0.02 |

## F COMPARISON WITH LIGHTWEIGHT GNNS

We further compare with UGS (Chen et al., 2021a), (Liu et al., 2022) and (You et al., 2021) on Cora and CiteSeer in the following table. The results show that our proposed method outperforms these three baselines in terms of classification accuracy with the same level of model parameters.

## G DEFEND AGAINST ADVERSARIAL ATTACKS

By incorporating the graph data into the optimization process, our method can effectively handle noisy data or data that has been manipulated by attackers. Specifically, the curriculum graph sparsification allows GASSIP to filter out edges that are either noisy or have been added maliciously by attackers. As a result, our approach exhibits a degree of robustness in the face of such adversarial scenarios.

Table 7: Comparison with lightweight GNNs

| Method | UGS (Chen et al., 2021a) | (Liu et al., 2022) | (You et al., 2021) | GASSIP |
|---|---|---|---|---|
| Cora | 80.30 | 81.88 | 82.83 | **83.20** |
| CiteSeer | 70.40 | 71.23 | 70.48 | **71.41** |

**Noisy Data.** We compare our methods with various baselines such as graph sparsification methods like DropEgde (Rong et al., 2020), PTDNet (Luo et al., 2021), and NeuralSparse (Zheng et al., 2020) on noisy Cora data by randomly adding 1k/5k edges in Table 8. This result demonstrates that when there exist noisy edges, GASSIP could achieve the best performance compared with baselines.

Table 8: Results under noisy edges.

| # noisy edge | GCN | GAT | DropEdge | PTDNet | NeuralSparse | GraphNAS | GASSO | DARTS | UGS | GASSIP |
|---|---|---|---|---|---|---|---|---|---|---|
| 1k | 78.37±0.47 | 77.99±0.75 | 78.16±0.44 | 77.16±1.27 | 79.24±0.51 | 79.14±0.46 | 78.73±0.90 | 78.57±0.63 | 78.65±0.53 | **79.26±0.49** |
| 5k | 69.14±0.55 | 67.42±0.74 | 68.28±0.63 | 66.48±1.23 | 68.68±0.53 | 71.28±0.59 | 70.76±0.95 | 71.76±0.88 | 69.49±0.62 | **73.80±0.64** |

**Poisoning Data.** To prove the defensive potential of our joint data and architecture optimization algorithm in countering adversarial attacks, we conduct experiments on perturbed data including comparison with state-of-the-art defensive methods like GCN-Jaccard (Wu et al., 2019) and RGCN (Zhu et al., 2019). Table 9 demonstrates that GASSIP exhibits defensive abilities against perturbed data. However, in order to further enhance its robustness against adversaries, it is necessary to develop special designs tailored to attack settings. Despite this, the current experiment clearly illustrates that incorporating data into the optimization objective function has the potential to alleviate the detrimental effects caused by adversaries.

Table 9: Denfensive performance under non-targeted attack (Mettack (Zügner & Günnemann, 2019)).

| Dataset | GCN | GAT | Arma | DropEdge | GCN-Jaccard | RGCN | GraphNAS | GASSO | DARTS | **GASSIP** |
|---|---|---|---|---|---|---|---|---|---|---|
| Cora | 66.93±1.06 | 68.61±2.24 | 65.09±1.28 | 68.48±1.44 | 69.89±1.19 | 67.20±1.02 | 70.05±1.27 | 66.51±2.76 | 61.05±1.32 | **73.05±0.69** |
| CiteSeer | 56.20±1.46 | 62.31±1.46 | 60.11±1.30 | 56.16±1.74 | 56.97±1.90 | 57.40±0.96 | 62.09±3.41 | 57.88±2.09 | 61.59±1.13 | **65.52±0.45** |

