# OpenReview forum: "Lightweight Graph Neural Network Search with Graph Sparsification"
_ICLR.cc/2024/Conference — Submitted to ICLR 2024_

### Official Review · Reviewer_hwTX · 2023-10-29

**Soundness:** 2 fair
**Presentation:** 3 good
**Contribution:** 2 fair
**Rating:** 3
**Confidence:** 5

**Summary:**

The paper introduces a novel approach called GASSIP (Lightweight Graph Neural Architecture Search with Graph Sparsification and Network Pruning) for automating the design of efficient Graph Neural Networks (GNNs). It highlights the needs for lightweight GNAS and data sparsification to reduce resource requirements. GASSIP employs operation pruning and curriculum graph data sparsification to iteratively optimize GNN architectures and graph data, resulting in more efficient and accurate lightweight GNNs. Experimental results demonstrate its superiority over traditional GNNs and GNAS, achieving substantial performance improvements with significantly reduced search time.

**Strengths:**

1.	GASSIP is shown to significantly improve the efficiency of GNAS by reducing search time while maintaining or even enhancing the performance of GNNs.
2.	The writing is easy to follow.

**Weaknesses:**

1.	The research problem is not novel to the community. There have been a series of work for the co-optimization of neural architecture and data, even at the domain of graph NAS and graph data.
2.	I am still concerned about the motivation to work on graph NAS. Different to other neural architectures, there are only several layers in GNNs, and the number of candidate operations is limited. I believe if one only applies the popular toolkit of hyperparameter tuning, the much higher performance will be obtained.
3.	The unstructured pruning of model weights makes no sense in the practical efficiency improvement. Based on the current parallel hardware (with processing of single instruction multiple data), the unstructured matrix multiplication has almost the same cost with the dense matrix multiplication.

**Questions:**

1.	Please address my concerns listed in the weaknesses.
2.	It is unscalable to apply a learnable mask with shape of N\timesN in graph data. The node number in most of the graphs are at the scale of millions or even billions.
3.	Following the last question, I need to check the possibility of applying this work in the benchmark datasets of ogbn-products and paper100m.
4.	Graph sparsification is a very old topic, and there have been many researches being conducted to provide the principle in how to remove edges without affecting the graph structural properties (e.g., adjacency eigenvalues). For example, one can remove edges based on degrees of 1/d_i + 1/d_j, where d_i and d_j are the degrees of node i and j, respectively. It is easy to remove more than 90% of edges but maintain the comparable performance [1].
[1] Lovász, László. "Random walks on graphs." Combinatorics, Paul erdos is eighty 2.1-46 (1993): 4
5.   How many edges can be deleted in the adopted datasets?

---

> ### Author Response · Authors · 2023-11-21
> **Response to reviewer hwTX (1/2)**
>
> Thank you for your feedback and the raised questions. We are happy that you found our proposed approach GASSIP is novel and our effort to improve the efficiency of GNAS is significant. We are glad to address the concerns you mentioned:
>
> **Q1: The research problem of the co-optimization of neural architecture and data, even in the domain of graph NAS and graph data is not novel.**
>
> A1: Thank you for your opinion. **On one hand**, co-optimization of the neural architecture and graph structure is essential to empowering the GNAS ability, given the discrete space of graph structure makes the traditional optimization that solely considers NAS achieve unsatisfactory performance when applied to GNNs [1]. Therefore, the problem of how to co-optimize GNAS with the underlying graph structure is still not fully solved. **On the other hand**, as we mentioned in the introduction, the main purpose of this paper is to search for lightweight GNN (i.e., lightweight GNN design), which is the first work to search for a lightweight GNN design while considering sparsifying the graph structure simultaneously. In order to select useful information in the graph structure to help search, we propose a novel graph learning algorithm (by co-optimizing the graph structure in Eq. (3)) as you pointed out.
>
> **Q2: The motivation to work on graph NAS.**
>
> A2: Thank you for the comment, though we respectfully disagree with this opinion. It is non-trivial to combine the strength of AutoML and graph machine learning given the following challenges [1-2]: 1) Unlike audio, image, or text, which has a grid structure, graph data lies in a non-Euclidean space. Thus, graph machine learning usually has unique architectures and designs (e.g., neighborhood aggregation functions, pooling functions, and edge dropout strategies ) and could also be constructed very deep [3-4]. For example, typical NAS methods focus on the search space for convolution and recurrent operations, which is distinct from the building blocks of GNNs. 2) Graph tasks per se are complex and diverse, ranging from node-level to graph-level problems, and with different settings, objectives, and constraints. How to impose proper inductive bias and integrate domain knowledge into a graph AutoML method is indispensable. 3) As the results are shown in Table 3 of AutoGL [5], naively applying HPO tuning methods could not surpass the performance of the specifically designed GNAS methods. Therefore, if one can simply use a hyperparameter tuning toolkit on GNNs without considering the GNN architecture to obtain SOTA performance, that could be remarkable progress in this research community.
>
> **Q3: The unstructured pruning of model weights makes no sense in the practical efficiency improvement.**
>
> A3: Thank you for pointing out this concern. Pruning with a mask is a common approach in literature[6]. In practice, the pruning mask becomes quite sparse after several iterations, therefore the pruning is mostly sparse matrix multiplication, which is more efficient compared with dense matrix multiplication. We have included this observation in the revision.
>
> **Q4: It is unscalable to apply a learnable mask with the shape of N\timesN in graph data.**
>
> A4: Sorry for this confusion. In practice, we use sparse matrix-based implementation, which means that our learnable mask is $|E|$ and is applied on the edge_index which has shape $|E| \times 2$, where $|E|$ is the number of edges in the graph. Given the graph is naturally sparse in practice, therefore it is not the bottleneck for extending to graphs of billions of nodes. We have improved this detail in our revision.
>
> **Q5: The possibility of applying this work in the benchmark datasets of ogbn-products and paper100m.**
>
> A5: Thank you for this concern. For the issue of large-scale datasets, please kindly refer to the reply to the common question. In short, it is possible to apply this work in large-scale graphs by combining our current implementation with a sampling technique. However, naively applying sampling techniques to the current supernet training in GNAS will result in consistency collapse issues [7]. Since our main focus is to search for lightweight GNNs with limited computational resources, we would like to leave this design of scalable and optimized sampling strategy as a future work to extend GASSIP to graph with a much larger scale.

---

> ### Author Response · Authors · 2023-11-21
> **Response to reviewer hwTX (2/2)**
>
> **Q6: Graph sparsification is a very old topic.**
>
> A6: Thank you for this opinion and the example. We agree that the problem of graph sparsification is a long-standing research problem. However, it becomes unclear how the performance would be when a new method is applied to the sparsified graph structure. As shown in Figure 4 in the Appendix of our manuscript, the performance varies quite differently when the sparsity of the graph differs. Therefore, it is non-trivial to adopt a GNN architecture to a sparsified graph while maintaining its performance, let alone optimizing the target GNN to be lightweight simultaneously.
>
> Another illustration of your example that removes edges based on degrees could fail under a new setting is the adversarial attack on GNNs [8], where the degree-based method is a usually adopted baseline but has fair performance. This indicates that the problem could evolve therefore the research on graph sparsification still needs to be explored to tackle the new challenges.
>
> **Q7: How many edges can be deleted in the adopted datasets?**
>
> A7: Thank you for this question. The number of edges that are deleted in the adopted graphs is controlled by a hyperparameter $p%$, and it would be as low as 5% in the experiments.
>
> [1] Automated Machine Learning on Graphs: A Survey, IJCAI 2021
>
> [2] Graph Neural Architecture Search: A Survey, TST 2022
>
> [3] DeepGCNs: Making GCNs Go as Deep as CNNs, TPAMI 2021
>
> [4] Training Graph Neural Networks with 1000 Layers, ICML 2021
>
> [5] AutoGL: A Library for Automated Graph Learning, ICLR 2021 GTRL workshop
>
> [6] A Unified Lottery Ticket Hypothesis for Graph Neural Networks, ICML 2021
>
> [7] Large-scale graph neural architecture search, ICML 2022
>
> [8] Adversarial Attacks on Node Embeddings via Graph Poisoning, ICML 2019

---

> > ### Author Response · Authors · 2023-11-23
> > **Kindly requesting response from reviewer hwTX**
> >
> > Dear Reviewer hwTX:
> >
> > We deeply appreciate your insightful comments and questions, which have significantly contributed to improve the quality of our manuscript. We look forward to hearing from you whether our answers, additional results, and modifications to the manuscript have satisfactorily addressed your concerns. We welcome any further feedback and we are happy to promptly make any further improvements to our submission.

---

### Official Review · Reviewer_CX38 · 2023-10-30

**Soundness:** 3 good
**Presentation:** 3 good
**Contribution:** 3 good
**Rating:** 8
**Confidence:** 2

**Summary:**

This paper introduces a new approach to lightweight graph neural network architecture search called GASSIP. What sets it apart is that during the search process, it jointly considers graph data sparsification and operation pruning, allowing the discovered sub-architectures to achieve better performance with fewer parameters. It also exhibits a degree of robustness. Ultimately, it yields both sparse graphs and lightweight sub-architectures, enhancing search efficiency.

**Strengths:**

1. First jointly considers operation pruning and graph data sparsification in graph neural architecture search, which can efficiently search lightweight GNNs.
2. Uses curriculum learning strategy in graph data sparsification, which can more accurately identify redundant edges and obtain a sparse graph structure beneficial for downstream tasks

**Weaknesses:**

1. Considers graph sparsification and operation pruning at the same time, but does not provide a theoretical analysis of whether this iterative optimization converges.
2. Insufficient experiments on large-scale graph data: The large-scale graph data set used in the experiments of this article only contains OGBN-ARXIV. Therefore, more experiments on large-scale graph data are needed to verify the performance of the GASSIP method.

**Questions:**

1. Is the search result sensitive to the choice of random seed?
2. Simultaneous optimization of operation pruning and graph data sparsification may interfere with each other and lead to performance degradation. Could you provide some theoretical analysis on the convergence of the joint optimization process?
3. In equation (3), operation pruning and graph sparsification are combined with a logical OR operation. What is the rationale behind this design choice?

---

> ### Author Response · Authors · 2023-11-21
> **Response to reviewer CX38**
>
> We sincerely thank the reviewer for your detailed comments and insightful questions. We are happy to address the concerns you mentioned:
>
> **Q1: Theoretical convergence analysis.**
>
> A1: Thank you for pointing out this weakness. We admit that it is difficult to conduct a theoretical analysis on the convergence of our iterative optimization algorithm without proper assumptions as the convergence of single graph sparsification proved in [1]. Therefore, we include this as one limitation of our work as well as a future direction. On the other hand, we found that GASSIP converges well during practice.  Table 2 in Section 5.3 provides **searching time comparison** in comparison with other baselines. We could see that our proposed method is far faster than DARTS+UGS (a first-search-then-prune method that disentangles the co-optimization method by first searching architectures and then conducting network pruning and graph data sparsification). This indicates that both simultaneous optimization of operation pruning and graph data sparsification and first-search-then-prune show satisfactory convergence behavior in practice.
>
> **Q2: Is the search result sensitive to the choice of random seed?**
>
> A2: Thank you for this question. Here are the experimental results for GASSIP on node classification that is averaged for 100 runs (mean±std) using different random seeds:
>
> | Method   | Cora | CiteSeer | PubMed | Physics | Ogbn-Arxiv |
>
> | GASSIP | 83.20±0.42 | 71.41±0.57 | 79.50±0.30 | 98.46±0.06 | 71.30±0.23 |
>
> We can observe that the stds are relatively small, therefore the searched result is not sensitive to the choice of random seed. We have added this observation to the experimental results.
>
> **Q3: In equation (3), operation pruning and graph sparsification are combined with a logical OR operation. What is the rationale behind this design choice?**
>
> A3: We are sorry for this confusion. The symbol denotes the element-wise product operation. We have updated our manuscript to reflect this accordingly.
>
> **Q4: Experiments on large-scale graphs.**
>
> A4: Thank you for this concern. For the issue of large-scale datasets, please kindly refer to the reply to the common question. Since it requires calculating the output of every operation within the supernet on the full graph to estimate reliable architecture parameters, it is difficult for most of the GNAS methods to scale to Ogbn-product and Ogbn-Paper100M without a specifically designed sampling strategy.
>
> [1] Graph Sparsification by Universal Greedy Algorithms, Journal of Computational Mathematics 2023

---

### Official Review · Reviewer_WS6Y · 2023-10-30

**Soundness:** 3 good
**Presentation:** 3 good
**Contribution:** 2 fair
**Rating:** 5
**Confidence:** 4

**Summary:**

The paper proposed an lightweight GNAS algorithm. It iteratively optimizes graph data and architecture through curriculum graph sparsification and operation-pruned architecture search.

**Strengths:**

1. The paper proposes a GNAS method with graph sparification, which is an interesting exploration.
2. The paper applies the method to extensive experiments and shows good results on multiple datasets.

**Weaknesses:**

1. GNAS methods nowadays have been expanded to large-scale datasets, while the paper only showed the results on Physics and Ogbn-Arxiv datasets. Could you please give a more overall perfomance comparison with other GNAS methods like GUASS  on large-scale OGB datasets?

   [1] Large-scale graph neural architecture search, ICML 2022.

2. More comparsion on the unified benchmark will be appreciated, such as NAS-Bench-Graph.

   [2] Benchmarking Graph Neural Architecture Search, NIPS 2022.

3. Cited works in Section 2 are mostly before 2022 and the methods compared in Table 1 are all before 2021 which make the work out-dated. I do know there were multiple GNAS and graph sparsification methods proposed in 2022/2023. Maybe more cutting-edge research work as well as comparisions should be added.

4. The first contribution is proprosing a operation-pruned search method with learnable weight mask. However, the work of HM-NAS introduced this hierarchical masking on redundant operations, edges, and even the weights of supernet. It seems like transfering the idea on graphs. Maybe you should add this work and discuss the noble part of the first contribution compared with the learnable weight mask idea on edges in HM-NAS.

   [3] HM-NAS: Efficient Neural Architecture Search via Hierarchical Masking, ICCV 2019.

6. I feel a little confused about the workflow in Figure 1. I understand that the iterative training process of structure mask is between the gradient update of operations and architectures. However, the training process probably need to point back to the architecture searching part to illustrate the interactive training not directly getting the final sparsed graph and connecting to the pruned architecture. Maybe the training part and the procedures after binarizing masks can be seperated.

**Questions:**

See weakness.

---

> ### Author Response · Authors · 2023-11-21
> **Response to reviewer WS6Y**
>
> Thank you for your time and detailed feedback. We are happy to adapt our work to answer your questions and include all your suggested changes.
>
> **Q1. Comparison with GUASS**
>
> A1: Thank you for this question. For the issue of large-scale datasets, please kindly refer to the reply to the common question. Since it requires calculating the output of every operation within the supernet on the full graph to estimate reliable architecture parameters, it is difficult for most of the GNAS methods to scale to Ogbn-product and Ogbn-Paper100M without a specifically designed sampling strategy. We, therefore, report the results compared with GUASS on the scalable benchmarks as below:
>
> | Method   | Cora | CiteSeer | PubMed | Physics | Ogbn-Arxiv |
>
> | GUASS  | 82.05±0.21 | 70.80±0.41 | 79.48±0.16 | 96.76±0.08 | 71.85±0.41 |
>
> | GASSIP | 83.20±0.42 | 71.41±0.57 | 79.50±0.30 | 98.46±0.06 | 71.30±0.23 |
>
> We can find that GASSIP achieves better performance than GUASS on smaller graphs, but GUASS could handle graphs with more nodes and edges (Ogbn-Arxiv) as it is specially developed for large-scale datasets. However, our trained model is more lightweight and therefore can be applied in practical scenarios where computational resources are limited, which is not applicable to GUASS.
>
> **Q2: Comparision with NAS-Bench-Graph.**
>
> A2: Thank you for bringing up this suggestion. However, NAS-Bench-Graph only designs general search space without considering the pruning on either the graph structure or the GNN architecture. As a result, directly applying GASSIP on the NAS-Bench-Graph is inappropriate and induces less meaningful comparison. We appreciate your opinion and would leave the efforts to develop a unified benchmark for lightweight GNAS as future work.
>
> **Q3: Maybe more cutting-edge research work as well as comparisons should be added.**
>
> A3: Thank you for this suggestion. We have done our best to conduct supplementary research on the GNAS and graph sparsification methods proposed in 2022/2023. We have included them [1-4] in the Related Work section with the corresponding discussion. We have also included GUASS as an additional baseline as you suggested. Please provide any missing related work that you imply and we will also include them in the discussion. Nevertheless, to the best of our knowledge, our proposed GASSIP is the first work to search for a lightweight GNN design while considering sparsifying the graph structure simultaneously.
>
> **Q4: Difference with HM-NAS.**
>
> A4: Thank you for the information about HM-NAS and we are sorry for not being aware of this wonderful work before. We have added the corresponding discussion with HM-NAS in the related works. Specifically, HM-NAS aims to improve the architecture search performance by loosening the hand-designed heuristics constraint with three hierarchical masks on operations, edges, and network weights. In contrast, our focus is different from HM-NAS as we aim to search for a lightweight GNN considering co-optimizing the graph structure. To achieve this lightweight goal, a mask for network weight is naturally introduced, which is commonly used for network pruning.
>
> **Q5: The workflow in Figure 1.**
>
> A5: Thanks for your suggestion and we are sorry for the confusion. The "sparsified graph" mentioned in the original figure refers to the masked graph rather than the final binarized sparsed graph. The arrow is not directly connecting to the pruned architecture but connecting two different iterative parts. We adjust the arrows to make this clear. After revision, we do not include the final mask binarizing step which is illustrated in Line 7 in Algorithm 2. Figure 1 only displays the iterative training process which is shown in Lines 1-6 in Algorithm 2. We have also updated the caption of Figure 1 to make this illustration clear.
>
> [1] DFG-NAS: Deep and Flexible Graph Neural Architecture Search, ICML 2022
>
> [2] Do Not Train It: A Linear Neural Architecture Search of Graph Neural Networks, ICML 2023
>
> [3] Ricci Curvature-Based Graph Sparsification for Continual Graph Representation Learning, TNNLS 2023
>
> [4] DSpar: An Embarrassingly Simple Strategy for Efficient GNN Training and Inference via Degree-Based Sparsification, TMLR 2023

---

> ### Comment · Reviewer_WS6Y · 2023-11-22
> **Response to Authors**
>
> Dear Authors,
>
> Thanks for your feedback. The response to the scalability concern is not convincing for me. The authors argued that the extension to heavy-weight scenarios with very large-scale graphs is not the main focus of our work. In my view, large-scale graphs have not direct relationship with the meaning of ‘heavy-weight’. ‘Light-weight’ is reflected in the complexity of GNN. Instead, search a lightweight GNN for large-scale graphs could also be considered in the paper. Intuitively, both graph sparsification and operation pruning help to improve the efficiency of searching a lightweight GNN for large-scale graphs. Maybe, joint optimization of graph sparsification and GNN NAS is not a scalable solution.
>
> Considering this concern, I keep my rating unchanged.

---

> > ### Author Response · Authors · 2023-11-22
> > **Thank you for the response**
> >
> > Dear Reviewer WS6Y:
> >
> > Thank you for your reply and we appreciate your feedback. We are sorry for any confusion raised in our rebuttal for the scalability concern. By the term "heavyweight", we denote the large-scale graph that involves billions of nodes and adequate computational resources. There is another line of research focusing on the scalable GNNs that meet this demand by incorporating additional pre-processing and post-processing stages[1], developing a system with a search engine[2], and co-designing the GNN algorithm and system [3], etc. You are right about our motivation to develop lightweight GNNs with low complexity, and we do not claim that our main focus is to propose scalable GNNs. We will update the revision to add a discussion with these scalable GNNs in the related works soon.
> >
> > We agree with you on the advantages of graph sparsification and operation pruning in searching for a lightweight GNN for large-scale graphs. It is indeed possible to apply this work in large-scale graphs by combining our current implementation with a sampling technique. We have tried an experimental attempt on the Ogbn-Product dataset (millions of nodes rather than billions of nodes though) with a widely used neighbor sampling [4], and the result is indeed weaker than GUASS (79.65 vs. 81.26), which is as expected since naively applying sampling techniques to the current supernet training in GNAS will result in consistency collapse issues [5]. Therefore, we would like to leave the design of scalable and optimized sampling strategy and the scalable choices of search space as future work to extend GASSIP to graph with a much larger scale.
> >
> > Many thanks again for your reply and we are happy to discuss more on your further concerns.
> >
> >
> > [1] Node Dependent Local Smoothing for Scalable Graph Learning, NeurIPS 2021
> >
> > [2] PaSca: A Graph Neural Architecture Search System under the Scalable Paradigm, WWW 2022
> >
> > [3] Algorithm and System Co-design for Efficient Subgraph-based Graph Representation Learning, VLDB 2022
> >
> > [4] https://pytorch-geometric.readthedocs.io/en/latest/_modules/torch_geometric/loader/neighbor_loader.html
> >
> > [5] Large-scale graph neural architecture search, ICML 2022

---

### Author Response · Authors · 2023-11-21
**Response to the Common Question**

We thank all of the reviewers for their thoughtful feedback and recognition of our paper’s contributions. We have addressed the common question here, individual reviewers’ comments in their replies, as well as updated the manuscript PDF with the changes suggested. All changes in the PDF are marked in pink for ease of reference.

**Common Question: Experiments on large-scale graphs with massive nodes such as Ognb-products or Ogbn-Paper100M**

We would like to thank all three reviewers for their curiosity about the performance of our method on very large-scale graphs such as Ognb-products or Ogbn-Paper100M.

On one hand, we admit that the current implementation of the GNAS-based methods (such as the SOTA GNAS method [1-3]) could barely be extended to graphs at such a scale without a specifically designed sampling strategy (except [4] as pointed out by Review WS6Y). Indeed, this is a common obstacle for this line of research since naively applying sampling techniques to the current supernet training in GNAS will result in consistency collapse issues [4]. This problem becomes more troublesome under our setting where the graph sparsification also needs to be optimized since the joint optimization of graph sparsification and large-scale graphs is also not well studied [5].

On the other hand, we kindly clarify that the extension to heavyweight scenarios with very large-scale graphs is not the main focus of our work. As we mentioned in the introduction, the main purpose of this paper is to search for a lightweight GNN (i.e., lightweight GNN design) that offers a wider range of application scenarios (e.g. edge computing) by limited computational resource requirements. We utilize graph structure to help identify effective optimal sub-architectures to reach the lightweight GNN goal, which is one of our main contributions.

In conclusion, we value the question of how to integrate our method with graphs with billions of nodes, but we aim to tackle the practical situation where the computational resources are limited. Note that this difficulty of scalability commonly hinders both graph sparsification and GNAS research in applications with contained resources. Therefore, we revised our manuscript to reflect this limitation and chose to leave this as a future work. Meanwhile, we hope the reviewers show solicitude to our main contributions as mentioned in the introduction.

[1] DFG-NAS: Deep and Flexible Graph Neural Architecture Search, ICML 2022

[2] Do Not Train It: A Linear Neural Architecture Search of Graph Neural Networks, ICML 2023.

[3] Graph differentiable architecture search with structure learning, NeurIPS 2021

[4] Large-scale graph neural architecture search, ICML 2022

[5] Survey on Graph Neural Network Acceleration: An Algorithmic Perspective, IJCAI 22

---

### Meta-Review · Area_Chair_yXyj · 2023-12-15

**Metareview:**

The paper addresses a gap in the existing research on Graph Neural Architecture Search (GNAS) by focusing on its applications in resource-constrained scenarios. Introducing a joint mechanism for graph data and architecture, the proposed method, Lightweight Graph Neural Architecture Search with Graph Sparsification and Network Pruning (GASSIP), identifies crucial sub-architectures using valuable graph data. GASSIP incorporates an operation-pruned architecture search module for efficient lightweight Graph Neural Network (GNN) search, and a curriculum graph data sparsification module with an architecture-aware edge-removing difficulty measurement to select optimal sub-architectures. The iterative optimization of these two modules, facilitated by differentiable masks, efficiently searches for the optimal lightweight architecture. Experimental results on five benchmarks demonstrate the effectiveness of GASSIP. Notably, the method achieves node classification performance comparable or superior to existing GNNs, with half or fewer model parameters, and a sparser graph.

The reviewers raise important points regarding the evaluation of the proposed method and suggest expanding the comparison to include more recent GNAS methods, especially on large-scale datasets such as OGB datasets, and considering benchmarks like NAS-Bench-Graph.

The citation and comparison with more recent works, including GUASS [1] and NAS-Bench-Graph [2], is recommended to ensure the paper reflects the latest advancements in the field. Additionally, acknowledging and discussing the contributions and distinctions compared to recent works like HM-NAS [3], which introduced hierarchical masking, would enrich the discussion on the proposed method.

The reviewers have also expressed confusion about the workflow in Figure 1, suggesting a clearer illustration of the iterative training process involving the structure mask, and separation of the training part from the procedures after binarizing masks for better clarity.

Addressing these suggestions would contribute to a more comprehensive evaluation and discussion of the proposed method in the context of recent advancements in GNAS and graph sparsification methods.

Overall, the reviewers tended to reject the paper. One reviewer scored the paper with an 8 but was unfortunately not available during the rebuttal period and also provided the least rigorous review. Hence, I have placed more weight on the reviewers leaning or advocating for rejecting the paper.

**Justification For Why Not Higher Score:**

Several weaknesses

**Justification For Why Not Lower Score:**

-

---

### Decision · Program_Chairs · 2024-01-16

Reject